# ATTENTION AS A COMPASS: EFFICIENT EXPLORATION FOR PROCESS-SUPERVISED RL IN REASONING MODELS

**Runze Liu**[1,2*], **Jiakang Wang**[2], **Yuling Shi**[3], **Zhihui Xie**[4], **Chenxin An**[4], **Kaiyan Zhang**[1,5],
**Jian Zhao**[1], **Xiaodong Gu**[3], **Lei Lin**[2], **Wenping Hu**[2], **Xiu Li**[1†], **Fuzheng Zhang**[2],
**Guorui Zhou**[2†], **Kun Gai**[2]
[1]Tsinghua University, [2]Kuaishou Technology, [3]Shanghai Jiao Tong University,
[4]The University of Hong Kong, [5]Frontis.AI

## ABSTRACT

Reinforcement Learning (RL) has shown remarkable success in enhancing the reasoning capabilities of Large Language Models (LLMs). Process-Supervised RL (PSRL) has emerged as a more effective paradigm compared to outcome-based RL. However, existing PSRL approaches suffer from limited exploration efficiency, both in terms of branching positions and sampling. In this paper, we introduce a novel PSRL framework (AttnRL), which enables efficient exploration for reasoning models. Motivated by preliminary observations that steps exhibiting high attention scores correlate with reasoning behaviors, we propose to branch from positions with high values. Furthermore, we develop an adaptive sampling strategy that accounts for problem difficulty and historical batch size, ensuring that the whole training batch maintains non-zero advantage values. To further improve sampling efficiency, we design a one-step off-policy training pipeline for PSRL. Extensive experiments on multiple challenging mathematical reasoning benchmarks demonstrate that our method consistently outperforms prior approaches in terms of performance and sampling and training efficiency. Our code is available at https://github.com/RyanLiu112/AttnRL.

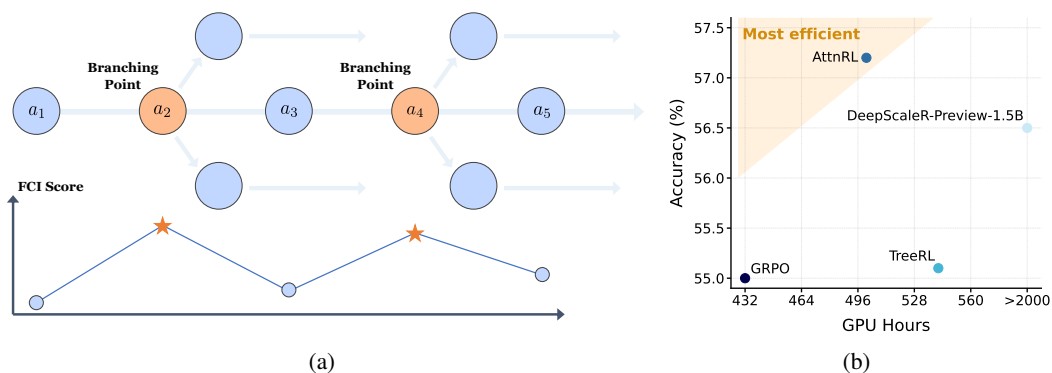

Figure 1: An Illustration of AttnRL. (a) AttnRL branches at steps with high attention scores. (b) AttnRL outperforms the baselines with great efficiency.

## 1 INTRODUCTION

Large Language Models (LLMs) have achieved remarkable progress in recent years (OpenAI, 2023; Hurst et al., 2024; Anthropic, 2023), particularly in their reasoning capabilities (OpenAI, 2024;

---

[*] Work done during an internship at Kuaishou Technology
[†] Corresponding authors: Xiu Li (li.xiu@sz.tsinghua.edu.cn), Guorui Zhou (zhouguorui@kuaishou.com)

Guo et al., 2025a). With the success of DeepSeek-R1 (Guo et al., 2025a), Reinforcement Learning with Verifiable Rewards (RLVR) has emerged as an effective post-training paradigm for further strengthening the reasoning abilities of LLMs (Shao et al., 2024; Zeng et al., 2025; Luo et al., 2025; Yu et al., 2025; Liu et al., 2025c; Hu et al., 2025; He et al., 2025a; An et al., 2025; Zhang et al., 2026; Wang et al., 2025; Zheng et al., 2025a).

Common RLVR approaches, such as Group Relative Policy Optimization (GRPO) (Shao et al., 2024) and its variants (Yu et al., 2025; Liu et al., 2025c; Yue et al., 2025), assign uniform training signals to all tokens within the same response, thereby overlooking fine-grained reasoning quality. In contrast, Process-Supervised RL (PSRL) methods refine credit assignment with Monte Carlo (MC) sampling to estimate step-level advantages (Hou et al., 2025; Guo et al., 2025b; Yang et al., 2025b; Zheng et al., 2025b; Li et al., 2025). However, existing PSRL methods suffer from several limitations: (1) they segment responses by fixed token length or entropy, ignoring the semantic meaning of model outputs; (2) they adopt uniform sampling across prompts and responses, leading to inefficient exploration; (3) they typically rely on two-step sampling per update, which significantly increases computational cost.

To overcome these limitations, we introduce **AttnRL**, a novel PSRL framework that improves both exploration and training efficiency. Our approach is motivated by the observation that attention scores serve as a meaningful metrics for identifying important reasoning behaviors in the model output. We therefore introduce an attention-based branching strategy for Monte Carlo sampling. To further enhance efficiency, we design an adaptive sampling mechanism that prioritizes difficult problems while filtering easier ones, and an adaptive batch sampling strategy that guarantees non-zero advantage values across batches. The experimental results on mathematical reasoning tasks demonstrate that AttnRL outperforms strong outcome-based and process-based baselines with great efficiency. An overview of AttnRL is shown in Figure 1.

The contributions of this work can be summarized as follows:

- We analyze the relationship between attention scores and reasoning behaviors, and propose attention-based branching method for PSRL.

- We develop an adaptive sampling mechanism that balances exploration across problems of varying difficulty and ensure valid training batches without zero advantage values.

- Empirical results on six mathematical benchmarks demonstrate the superiority of our method beyond the baselines in both performance and efficiency.

## 2 PRELIMINARIES

### 2.1 LLM REASONING AS A STEP-LEVEL MARKOV DECISION PROCESS

Following Sutton & Barto (2018); Zhang et al. (2025), we formulate LLM reasoning as a Markov Decision Process (MDP) defined by the tuple $(\mathcal{S}, \mathcal{A}, \mathcal{P}, \mathcal{R}, \gamma)$, where $\mathcal{S}$ is the state space, $\mathcal{A}$ is the action space, $\mathcal{P} : \mathcal{S} \times \mathcal{A} \mapsto \mathcal{S}$ is the transition dynamics, $\mathcal{R} : \mathcal{S} \times \mathcal{A} \mapsto \mathbb{R}$ is the reward function, and $\gamma \in [0, 1]$ is the discount factor. In the LLM setting with a prompt dataset $\mathcal{D}$, the initial state is $s_1 = q \sim \mathcal{D}$. The state transition is deterministic, since the next state is formed by concatenating the current state with the generated action: $s_{k+1} = [s_k, a_k]$, where $[\cdot, \cdot]$ denotes string concatenation. For process-level supervision of LLMs (Zhang et al., 2025; Liu et al., 2025b), actions are defined at the step level, where each action $a_t$ corresponds to a semantically coherent segment such as a sentence or a paragraph, rather than a single token. In this paper, we adopt this step-level MDP formulation.

### 2.2 OUTCOME-SUPERVISED AND PROCESS-SUPERVISED RL

**Outcome-Supervised RL.** Group Relative Policy Optimization (GRPO) (Shao et al., 2024) is an Outcome-Supervised RL (OSRL) method that eliminates the need for an explicit critic model by estimating the advantage using the rewards $\{R_1, \cdots, R_G\}$ of $G$ sampled rollouts $\{o_1, \cdots, o_G\}$. The normalized advantage is computed as $\hat{A}_{i,t} = \frac{R_i - \text{mean}(\{R_i\}_{i=1}^G)}{\text{std}(\{R_i\}_{i=1}^G)}$. The GRPO objective is then given

by:

$$\mathcal{J}_{\text{GRPO}}(\theta) = \mathbb{E}_{q \sim \mathcal{D}, \{o_i\}_{i=1}^G \sim \pi_{\theta_{\text{old}}}(\cdot|q)}$$

$$\left[ \frac{1}{G} \sum_{i=1}^G \frac{1}{|o_i|} \sum_{t=1}^{|o_i|} \left( \min \left( r_{i,t}(\theta) \hat{A}_{i,t}, \text{clip} \left( r_{i,t}(\theta), 1 - \varepsilon, 1 + \varepsilon \right) \hat{A}_{i,t} \right) - \beta \mathbb{D}_{\text{KL}}(\pi_\theta \| \pi_{\text{ref}}) \right) \right], \tag{1}$$

where $r_{i,t} = \frac{\pi_\theta(o_{i,t}|q, o_{i,<t})}{\pi_{\theta_{\text{old}}}(o_{i,t}|q, o_{i,<t})}$ is the importance sampling ratio, and $\beta$ controls the strength of the KL divergence penalty that regularizes the policy towards the reference policy $\pi_{\text{ref}}$.

**Process-Supervised RL.** For PSRL, the sampling process usually includes two stages: (1) **Initial Sampling**: Sample multiple responses to the problem; (2) **Monte Carlo Sampling**: Select several tokens as division points and rollout twice starting from these branching positions (Hou et al., 2025; Guo et al., 2025b; Yang et al., 2025b). In this paper, we follow the setting of TreeRL (Hou et al., 2025), which proposes a tree-based advantage estimation method. For each node, the value is computed as the average accuracy of its all children:

$$V(s_k) = \frac{1}{|L(s_k)|} \sum_{l \in L(s_k)} \mathbf{1}(l \text{ is correct}), \tag{2}$$

where $L(s_k)$ denotes the children of node $s_k$. The final advantage is the summation of global advantage ($V(s_k) - V(s_1)$) and local advantage ($V(s_k) - V(p(s_k))$):

$$\hat{A}_{i,k} = \frac{1}{\sqrt{|L(s_k)|}} \Big( \underbrace{V(s_k) - V(s_1)}_{\text{global advantage}} + \underbrace{V(s_k) - V(p(s_k))}_{\text{local advantage}} \Big), \tag{3}$$

where $\sqrt{|L(s_k)|}$ is used to reduce the optimization strength of the non-leaf steps to prevent overfitting (Hou et al., 2025) and $p(s_k)$ is the parent node of $s_k$. Then the policy is optimized using the loss function in equation 1, which is the same as that of OSRL but differs at the advantage granularity.

### 2.3 ATTENTION MECHANISM

Modern LLMs are typically decoder-only Transformer-based architectures (Vaswani et al., 2017; Yang et al., 2024; 2025a), and the core operation inside each Transformer block is the (masked) self-attention mechanism. For a given layer $l$ and head $h$, the model first computes query $Q^{l,h}$, key $K^{l,h}$ and value matrices. Then the attention score $\alpha$ is computed as:

$$\alpha^{l,h} = \text{softmax} \left( \frac{Q^{l,h} K^{l,h^\top}}{\sqrt{d_k}} + \text{mask} \right), \tag{4}$$

where $d_k$ is the per-head dimensionality. In vanilla causal attention, attention mask blocks access to all future tokens by assigning them $-\infty$, while past and current tokens remain unmasked with $0$.

## 3 METHOD

In this section, we present AttnRL, an exploration-efficient method for process-supervised RL. We begin by examining the role of massive attention values and how they can be leveraged to identify branching points via attention scores to explore at important branches (Section 3.1). To enable more efficient exploration, we propose an adaptive sampling strategy that avoids oversampling easy problems and ensures each training batch contains only samples with non-zero advantage (Section 3.2). Finally, we introduce our efficient training pipeline based on one-step off-policy learning (Section 3.3).

### 3.1 BRANCHING AT MASSIVE ATTENTION VALUES

Prior work has demonstrated that massive attention values in self-attention mechanisms play a critical role in contextual knowledge understanding (Jin et al., 2025), as they highlight tokens most relevant for answering questions. Motivated by this insight, we investigate two key questions: (1) Do massive attention values consistently appear in complex reasoning tasks? (2) What impact do these massive attention values have, and how can they be effectively utilized in RL training?

### 3.1.1 MASSIVE ATTENTION VALUES IN LLMs

**Step 1: Segmenting and computing step-level attention scores.** Following prior work on process supervision (Wang et al., 2024a; Liu et al., 2025b), we first segment the entire response into multiple steps using two consecutive line breaks ("\n\n"), yielding $T_k$ steps: $o = (o_1, o_2, \ldots, o_{T_k})$. Next, we extract token-to-token attention scores via a single forward pass. By aggregating these scores at the step level, we obtain step-to-step attention matrices $\alpha^{l,h} \in \mathbb{R}^{T_k \times T_k}$, where $\alpha^{l,h}_{j,k}$ denotes the attention weight of step $j$ attending to step $k$ at layer $l$ and head $h$.

**Step 2: Computing the Forward Context Influence (FCI) score.** To quantify the influence of a given step on subsequent tokens, we define the Forward Context Influence (FCI) score at layer $l$ and head $h$ by summing the attention scores over the subsequent steps:

$$y_k^{l,h} = \sum_{j=k+\Delta}^{T_k} \alpha_{j,k}^{l,h}, \tag{5}$$

where $\Delta$ is a hyperparameter that restricts the scope to sufficiently distant parts of the response, set to 4 following Bogdan et al. (2025). We then aggregate across layers and heads by taking the maximum value:

$$y_k = \max_{l,h} \{y_k^{l,h}\}. \tag{6}$$

The resulting FCI score $y_k$ captures the degree to which step $k$ influences the downstream context at the attention level. An illustrative visualization of steps with large FCI values is provided in Figure 2. From this figure, we can see that most steps with high FCI scores or peak FCI values are related to reasoning behaviors, such as planning and self-verification (Bogdan et al., 2025). The full response are listed in Table 5 in Appendix C.

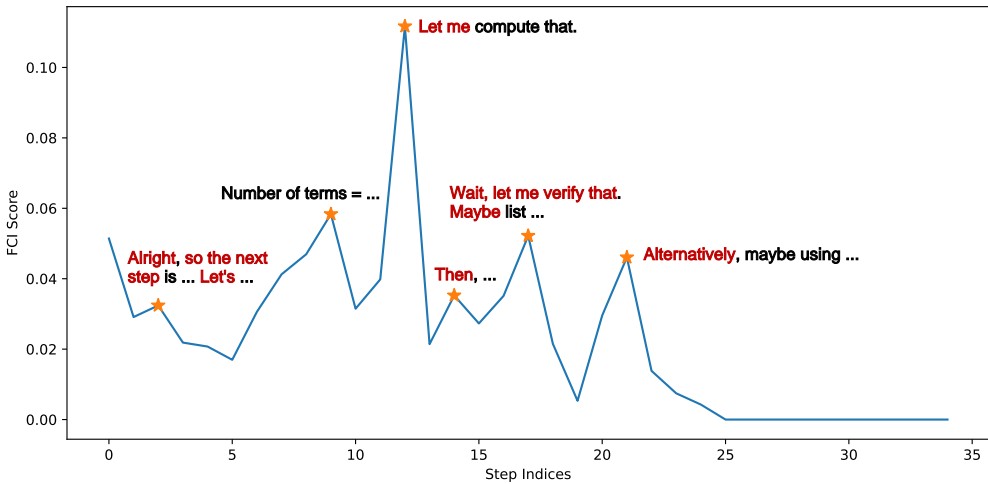

Figure 2: The visualization of steps with high FCI scores. The words in red denote reasoning behaviors.

### 3.1.2 THE EFFECTS OF STEPS WITH HIGH FCI SCORES

After identifying and qualitatively analyzing steps with high FCI scores, we conduct quantitative experiments to examine the impact of disrupting attention values on performance. Specifically, we select a step either (1) randomly from the top 20% of steps ranked by FCI scores (denoted as "Top 20%") or (2) randomly from the remaining steps (denoted as "20%–100%"). For the chosen step, we set its corresponding attention values to zero. We hypothesize that disrupting attention at key steps (with high FCI scores) will cause greater performance degradation compared to disrupting other steps. We test this hypothesis on AIME24 (MAA, 2024) using DS-R1-Distill-Qwen-1.5B, with each problem sampled four times. The results shown in Figure 3(a) confirm that disrupting top 20% steps leads to a significant drop in accuracy. Furthermore, we investigate the effect of disruption position. We divide the disruption positions relative to the original response length into five uniform bins. As shown in Figure 3(b), accuracy exhibits an increasing trend as the disruption position moves later in

the sequence, indicating that disruptions at earlier positions have a larger negative impact on final performance.

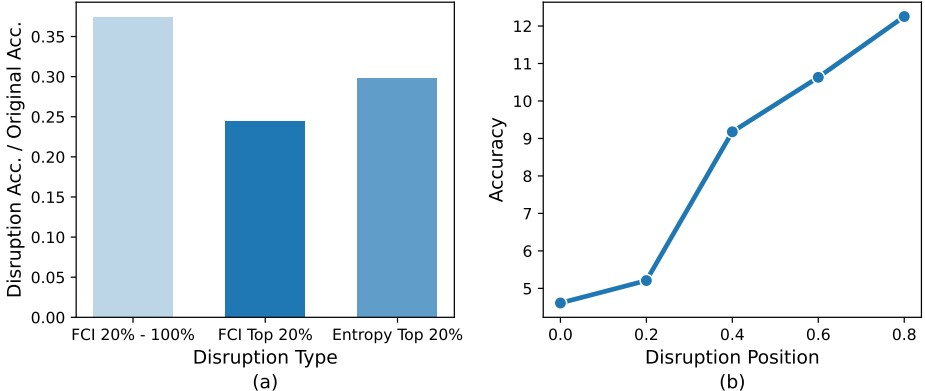

Figure 3: Disruption results on AIME24 and AIME25. (a) Normalized average accuracy of different disruption types. (b) Average accuracy of different disruption positions.

### 3.1.3 ATTENTION-BASED TREE BRANCHING

Based on the analysis in Section 3.1.1 and 3.1.2, we have identified that steps with high FCI scores are related to reasoning behaviors and have strong influences on the the reasoning performance. Now we propose Attention-based Tree Branching (ATB), which builds the branches of the tree at steps with high FCI scores.

Specifically, we compute the FCI score for each step using equation 6 after initial sampling to enable effective exploration. We then select the top 20% of the steps with the highest FCI scores for branching:

$$C = \{k \mid k \geq \text{Quantile}(y_1, \ldots, y_{T_k}, \rho)\}, \tag{7}$$

where $\rho = 0.2$ is the quantile level. However, randomly selecting steps with high FCI scores as branching points can be suboptimal, as misleading initial steps may lead the reasoning process in incorrect directions and we have found that earlier steps have more influence on the final result. Similar phenomenons have also been found in Wen et al. (2025), which identifies these as "Tunnel Vision". To mitigate this, we select the top $N$ ($N = 2$ following Hou et al. (2025)) earliest steps from $C$ as branching points, ensuring that diverse reasoning paths are explored through attention-based branching.

### 3.2 ADAPTIVE SAMPLING

### 3.2.1 DIFFICULTY-AWARE EXPLORATION

**Attention-based Filtering.** Previous PSRL approaches explore all problems uniformly (Hou et al., 2025), which is highly inefficient. In particular, problems that are easy (i.e., achieving an accuracy of 100% at initial sampling) have a high probability (about 70% - 80%, shown in Figure 7(a)) of being correct at both sampling stages, leading to limited learning opportunities.

To address this, we propose an attention-based filtering method to identify problems that are too easy to sample an incorrect response. We compute the average massive attention values for all problems in the DeepScaleR (Luo et al., 2025) dataset. As shown in Figure 4, we empirically find that, for problems whose initial samples are all correct, if they have lower attention values, they will tend to have zero advantage values, indicating that all samples are correct. Therefore, we filter out problems with low attention values and only retain those with attention values above the average attention values:

$$\mathcal{D}_{\text{MC}} = \{q \mid \frac{1}{G} \sum_{i=1}^{G} \frac{1}{T_{i,k}} \sum_{k=1}^{T_{i,k}} y_{i,k} \geq \text{mean value}\}, \tag{8}$$

where $y_{i,k}$ is the attention score for the $k$-th step in problem $i$.

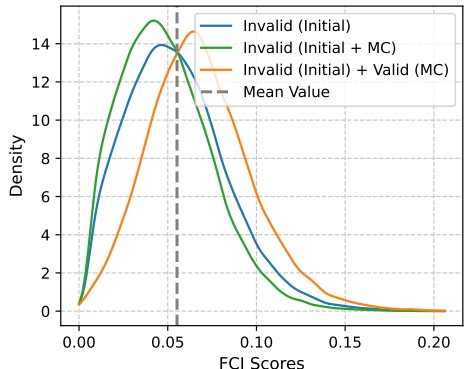

Figure 4: Average FCI scores of all problems during the training process of TreeRL on DeepScaleR dataset. "Invalid" means that the advantage is zero for all responses of that problem.

**Difficulty-aware Expansion.** After attention-based filtering, we expand different number of trees according to problem difficulty since it is more difficult to rollout correct responses for hard problems. Let the difficulty score be $z_n = \frac{1}{G}\sum_i \mathbf{1}(o_i \text{ is correct})$. Then the number of trees expanded for each problem $M$ is determined by the difficulty score:

$$M = \text{Round}(\exp(-z_n) \times M'), \tag{9}$$

where $\text{Round}(\cdot)$ denotes rounding to the nearest integer and $M'$ denotes original tree numbers and is set to 6 following Hou et al. (2025).

### 3.2.2 ADAPTIVE BATCH SAMPLING

After initial sampling and MC sampling, a large proportion of responses contribute nothing to training because their advantages are zero (detailed in Figure 7(b)). To ensure that each training batch remains effective, we introduce an adaptive batch size mechanism.

Let the target training batch size be $B'$, valid training batch size at step $m$ be $B''_m$, and the sampled prompt batch size at step $m$ be $B_m$. The sampling batch size at step $m$ is updated as:

$$B_{m+1} = \text{Round}(\lambda B_m + (1 - \lambda)\frac{B'}{B''_m}B_m), \tag{10}$$

where $\lambda$ is the weight balancing historical and current batch sizes. After MC sampling, responses with zero advantages are discarded, ensuring that all samples in the final batch have non-zero advantages, which improves training efficiency.

Our adaptive batch sampling differs from the dynamic sampling used in DAPO (Yu et al., 2025) in two key ways: (1) It requires only a single round of prompt sampling and generation per training step. (2) It avoids inefficiency from discarding valid responses when their number exceeds $B'$. As a result, the actual batch size naturally fluctuates around the target $B'$ while maintaining high training efficiency.

### 3.3 EFFICIENT TRAINING WITH ONE-STEP OFF-POLICY

Prior process-supervised RL methods typically require two sampling procedures per training iteration (Hou et al., 2025; Yang et al., 2025b; Guo et al., 2025b; Zheng et al., 2025b). This is highly inefficient, as sampling often dominates the overall training time. To address this, we propose a one-step off-policy learning framework for PSRL, inspired by recent advances in efficient RL training (Noukhovitch et al., 2025; Fu et al., 2025; Meituan-search, 2025).

In our approach, only a single sampling operation is performed at each training step. Concretely, at training step $m$, we conduct initial sampling for the $(m+1)$-th problem batch while simultaneously performing MC sampling for the $m$-th problem batch. This design ensures that the initial sampling for a batch occurs at step $m-1$, followed by its MC sampling at step $m$, thereby eliminating redundant sampling. As a result, the overall sampling cost is substantially reduced, leading to improved training efficiency. The full training pipeline of AttnRL is illustrated in Figure 5.

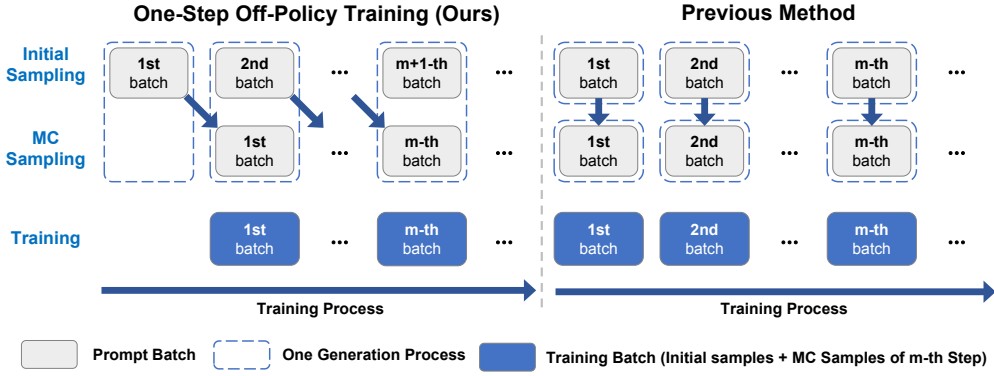

Figure 5: Training pipeline of AttnRL. Our method (left) only needs one-time generation per training iteration, while previous methods (right) require to sample twice and are inefficient.

## 4 EXPERIMENTS

### 4.1 SETUP

**Models and Baselines.** Following Hou et al. (2025), we adopt two supervised fine-tuned models, which are also reasoning models, as base models: DS-R1-Distill-Qwen-1.5B and DS-R1-Distill-Qwen-7B (Guo et al., 2025a). We compare against the following baselines: (1) **GRPO** (Shao et al., 2024): A representative OSRL method without critic model training. (2) **TreeRL** (Hou et al., 2025): The method is based on GRPO but differs that TreeRL samples with tree-based branching and estimates advantage values at segment-level. (3) **DeepScaleR-Preview-1.5B** (Luo et al., 2025): A strong RL-trained model with iterative context expansion at 1.5B scale.

**Evaluation and Metrics.** We evaluate all methods on six widely used mathematical reasoning benchmarks: AIME24 (MAA, 2024), AIME25 (MAA, 2025), AMC23 (MAA, 2023), MATH-500 (Lightman et al., 2024), Minerva Math (Lewkowycz et al., 2022), and OlympiadBench (He et al., 2024). We report both *Pass@1* and *Pass@K*, where $K = 32$ for AIME24, AIME25, and AMC23, and $K = 4$ for the remaining benchmarks. Evaluation is performed with a maximum response length of 32,768 tokens. For verification, we use a hybrid of DeepScaleR's verifier and Math-Verify[1] to ensure correctness (He et al., 2025a).

**Implementation Details.** We train all methods using DeepScaleR-Preview-Dataset following Luo et al. (2025); Liu et al. (2026), which contains 40.3k mathematical reasoning problems. We set the training batch size to 64, the PPO minibatch size to 32, and the learning rate to $1 \times 10^{-6}$. For all methods, we adopt token-level policy loss and apply Clip-Higher with $\varepsilon_{\text{high}} = 0.28$, following Yu et al. (2025). We use KL loss with weight 0.001 following Liu et al. (2025a); Wang et al. (2025). AttnRL is implemented based on TreeRL (Hou et al., 2025) and GRPO is used for policy optimization. We set $\lambda = 0.9$ (a standard EMA value (Kingma, 2014)) and $\rho = 0.2$.

The training is conducted using verl (Sheng et al., 2025), and rollouts are generated using vLLM (Kwon et al., 2023) with a maximum response length of 8,192 tokens, top-$p$ of 1.0, and temperature of 1.0 for both DS-R1-Distill-Qwen-1.5B and DS-R1-Distill-Qwen-7B. Experiments for DS-R1-Distill-Qwen-1.5B are conducted on a single node with $8\times$ NVIDIA H100 GPUs, and experiments for DS-R1-Distill-Qwen-7B are run on three nodes, each with $8\times$ NVIDIA H800 GPUs.

### 4.2 MAIN RESULTS

**AttnRL outperforms the base model.** As shown in Table 1, AttnRL outperforms the base model across all six benchmarks, achieving an average improvement of 7.5% for DS-R1-Distill-Qwen-1.5B.

---

[1] https://github.com/huggingface/Math-Verify

Table 1: Evaluation results on mathematical benchmarks. The results of AttnRL are shaded and the highest values are **bolded**.

| Method | AIME24 | AIME25 | AMC23 | MATH-500 | Minerva | Olympiad | Avg. |
|---|---|---|---|---|---|---|---|
| **DS-R1-Distill-Qwen-1.5B** | 28.3 | 23.0 | 71.8 | 84.8 | 35.6 | 54.9 | 49.7 |
| ↳ GRPO | 36.9 | 27.2 | 77.7 | 88.4 | 39.5 | 60.4 | 55.0 |
| ↳ DeepScaleR-Preview-1.5B | **40.5** | 28.3 | 81.0 | 89.5 | 38.1 | **61.8** | 56.5 |
| ↳ TreeRL | 36.7 | 27.1 | 78.9 | 88.5 | 38.7 | 60.9 | 55.1 |
| ↳ AttnRL | 39.7 | **28.5** | **83.2** | **90.0** | **40.3** | 61.4 | **57.2** |
| **DS-R1-Distill-Qwen-7B** | 54.0 | 40.0 | 89.8 | 94.1 | 48.1 | 70.0 | 66.0 |
| ↳ GRPO | 54.9 | 39.6 | 90.8 | 94.3 | 48.6 | 69.7 | 66.3 |
| ↳ TreeRL | 55.4 | 40.0 | 92.2 | 94.3 | 49.0 | 70.7 | 66.9 |
| ↳ AttnRL | **59.3** | **42.5** | **92.5** | **95.4** | **49.3** | **73.3** | **68.7** |

AttnRL surpasses the base model significantly on AIME24 benchmark, achieving an improvement of 11.4% and 5.3% for 1.5B and 7B models, respectively[2].

**AttnRL outperforms PSRL and strong RLVR baselines.** As reported in Table 1, AttnRL surpasses GRPO and TreeRL by an average of 1.9% and 1.8% across all benchmarks at 1.5B scale, confirming its effectiveness. Moreover, AttnRL outperforms DeepScaleR-Preview-1.5B, which is trained with a three-stage context extension (8K → 16K → 24K) over 1750 steps. In contrast, AttnRL achieves superior results with only 500 steps at an 8K response length, highlighting both its effectiveness and efficiency.

### 4.3 ABLATION STUDY

To evaluate the contribution of each component, we conduct an ablation study on the six mathematical benchmarks using DS-R1-Distill-Qwen-1.5B. As shown in Table 2, incorporating ATB alone improves performance over TreeRL by an average of 1.2%, while combining ATB with adaptive sampling allows AttnRL to achieve the highest performance. Importantly, filtering out problems whose responses are all correct after initial sampling results in a slight performance drop, as even "easy" problems can produce incorrect responses under Monte Carlo sampling, providing valuable training signals that enhance overall model performance.

Table 2: Results of ablation study on mathematical benchmarks. The results of AttnRL are shaded and the highest values are **bolded**.

| Method | AIME24 | AIME25 | AMC23 | MATH-500 | Minerva | Olympiad | Avg. |
|---|---|---|---|---|---|---|---|
| TreeRL | 36.7 | 27.1 | 78.9 | 88.5 | 38.7 | 60.9 | 55.1 |
| ↳ w/first 2 step branching | 35.6 | 28.9 | 79.5 | 89.4 | 38.7 | 60.5 | 55.4 |
| ↳ w/ATB | 39.1 | 27.2 | 81.4 | 89.2 | 40.1 | 61.0 | 56.3 |
| ↳ w/ATB + ADS (w/o attention-based filtering) | 38.4 | **29.1** | 81.0 | 89.8 | 38.7 | 61.2 | 56.4 |
| ↳ w/ATB + ADS (w/o difficulty-aware expansion) | 39.6 | 28.2 | 82.0 | **90.3** | 39.6 | 61.0 | 56.8 |
| ↳ AttnRL | **39.7** | 28.5 | **83.2** | 90.0 | **40.3** | **61.4** | **57.2** |

## 5 ANALYSIS

### 5.1 SAMPLING

**How does ATB outperform entropy-based tree branching?** The results in Table 2 show that TreeRL w/ATB outperforms TreeRL, which branches at tokens with highest entropy values. To further understand the effects of ATB, we plot four sampling curves during training process in Figure 6. For Figure 6(a) and (b), we visualize the solve all ratio (i.e., the ratio of problems whose outputs are all correct) and solve none ratio (i.e., the ratio of problems whose outputs are all wrong) of MC sampling, respectively. These two subfigures demonstrate that ATB enables more effective sampling at both easy and hard problems. Figure 6(c) and (d) show the valid ratio (i.e., the ratio of problems whose outputs are either not all correct nor all wrong) of MC sampling and both sampling, respectively. The results also demonstrate the effectiveness of ATB.

---

[2]The performance gain of 7B model is smaller than that of 1.5B model, which may because DeepScaleR-Preview-Dataset is originally used to fine-tune DS-R1-Distill-Qwen-1.5B and is relatively easy for DS-R1-Distill-Qwen-7B, which is stronger and has less room for improvement.

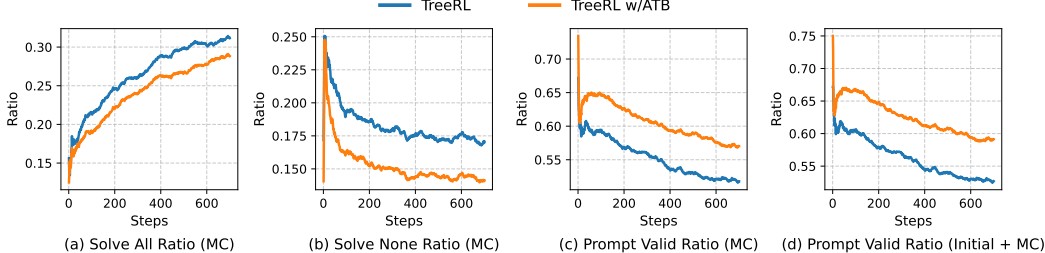

Figure 6: The sampling statistics of ATB and entropy-based branching. The curves are smoothed using EMA for better visualization.

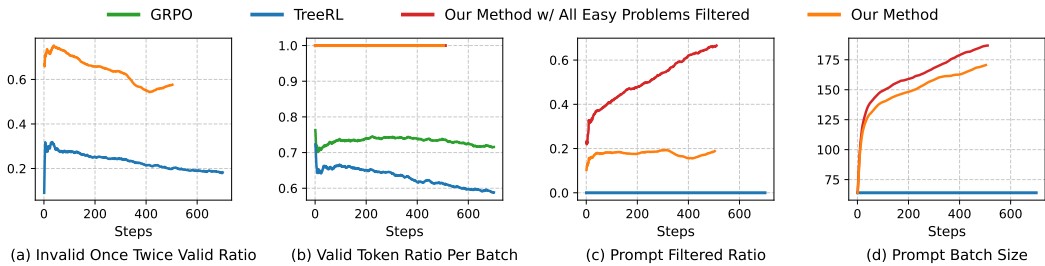

Figure 7: Curves related to sampling information statistics of all methods. The curves are smoothed using EMA for better visualization.

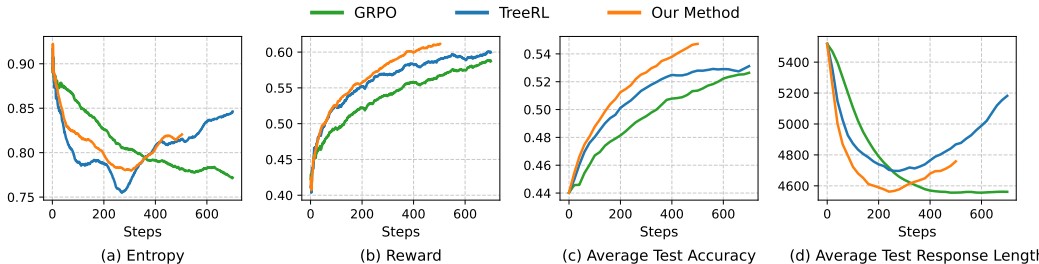

Figure 8: The training dynamics curves of all methods. The curves are smoothed using EMA for better visualization.

**Adaptive Sampling.** To better understand the effects of our proposed adaptive sampling method, we visualize the training curves related to the sampling process. The results in Figure 7(a) show that our method significantly reduces the ratio of both samples of two sampling steps are correct given the initial sampling results are correct, by filtering out prompts with low FCI scores (shown in Figure 7(c)). Additionally, AttnRL benefits from maintaining a valid training batch by dynamically adjust the prompt batch size (shown in Figure 7(d)), resulting in a training batch with all tokens having non-zero advantage values (shown in Figure 7(b)).

## 5.2 TRAINING DYNAMICS AND EFFICIENCY

**Training Dynamics.** The training dynamics of GRPO, TreeRL, and AttnRL are visualized in Figure 8. Figure 8(a) shows that the entropy curve of GRPO decreases along the training process, while PSRL methods first decreases then increases. Compared with TreeRL, AttnRL shows higher entropy, enabling more diverse exploration during training. Figure 8(b)-(c) show AttnRL learns faster with less training steps and Figure 8(d) shows the response length of AttnRL is shorter than that of TreeRL, demonstrating that AttnRL outperforms TreeRL at both final performance and reasoning conciseness.

**Training Efficiency.** As shown in Table 3, the training efficiency of the introduced one-step off-policy reduces the training time by 8% compared with original TreeRL implementation. AttnRL outperforms TreeRL with less wall-clock training time, more valid tokens for training (i.e., token with non-zero advantage values), and better overall performance significantly under the same computational resources. These strong efficiency improvements are achieved through especially at our adaptive sampling mechanism, which samples a dynamic batch of problems, filters out some low-value easy problems, and keeping a relatively stable size of batch with all samples useful for training.

Table 3: Comparison of training efficiency among AttnRL and baselines. The results of AttnRL are shaded and the best values are **bolded**.

| Method | Wall-clock Time | # Valid Training Tokens | Performance |
|---|---|---|---|
| GRPO | **54.0** | 656.0M | 55.0 |
| TreeRL | 67.7 | 274.6M | 55.1 |
| TreeRL w/one-step off-policy | 62.2 | 269.1M | 55.3 |
| AttnRL | 62.6 | **930.4M** | **57.2** |

## 6    RELATED WORK

### 6.1    REINFORCEMENT LEARNING FOR LLM

Reinforcement Learning has shown great success for enhancing the reasoning abilities of LLMs (OpenAI, 2024; Guo et al., 2025a). With the success of OpenAI o1 (OpenAI, 2024) and DeepSeek-R1 (Guo et al., 2025a), RLVR has become an efficient method for improving reasoning abilities of LLMs (Yu et al., 2025; Liu et al., 2025c; Chu et al., 2026; Yue et al., 2025; He et al., 2025a; Luo et al., 2025; Chen et al., 2025b; Liu et al., 2025a; Chen et al., 2025a; An et al., 2025; Wang et al., 2025; Zheng et al., 2025a). These works focus on outcome-based rewards that are inefficient for RL training, while our method focus on RL with process rewards.

### 6.2    PROCESS SUPERVISION FOR LLM

Process supervision has demonstrated superiority than outcome-based feedback in mathematical reasoning, especially Process Reward Models (PRMs) (Uesato et al., 2022; Lightman et al., 2024; Wang et al., 2024b). A line of works focus on token-level process rewards (Yuan et al., 2025; Cui et al., 2025; Fei et al., 2025), using DPO-like rewards (Rafailov et al., 2023; 2024) for policy learning. For PRM-based methods, a line of works (Wang et al., 2024b; Setlur et al., 2025; Cheng et al., 2025; Zha et al., 2025; Ye et al., 2025) use discriminative PRMs for RL training, while another line of works use generative PRMs (Zhao et al., 2026) to provide process rewards for RL training (Zou et al., 2025; He et al., 2025b; Xie et al., 2025). To mitigate reward hacking and avoid training an online PRM, some works use online Monte Carlo sampling to estimate process rewards (Kazemnejad et al., 2025; Hou et al., 2025; Guo et al., 2025b; Yang et al., 2025b; Zheng et al., 2025b; Li et al., 2025; Dong et al., 2026). Our method belong to the category which leveraging MC sampling to estimate process rewards. However, previous methods mainly focus on non-reasoning models and is inefficient from the perspective of both branching points, sampling mechanism, and two-step generation, while our work proposes effective and efficient methods of process supervision for reasoning models.

## 7    CONCLUSION

In this paper, we propose AttnRL for PSRL in reasoning models, which leverages attention information to find reasoning-related steps and branches at these positions for efficient exploration. Additionally, we introduce adaptive sampling based on problem difficulty and maintaining valid training batch size. Experimental results on mathematical reasoning benchmarks demonstrate the effectiveness and efficiency of our method.

## REPRODUCIBILITY STATEMENT

The implementation details of our method are discussed in Section 4 and Appendix A. The link of our code implementation is provided in Abstract.

ACKNOWLEDGMENTS

This work was supported by the STI 2030-Major Projects under Grant 2021ZD0201404 and Shenzhen Key Laboratory of New Generation Interactive Media Technology Innovation (ZDSYS20210623092001004).

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

## A    EXPERIMENTAL DETAILS

**Evaluation.**    For evaluation, we use the prompt listed in Table 4, following Luo et al. (2025).

Table 4: Prompt Template.

> {problem} Let's think step by step and output the final answer within `\boxed{}`.

## B    ADDITIONAL EXPERIMENTAL RESULTS

### B.1    FULL TEST CURVES

The test curves of six mathematical benchmarks are shown in Figure 9.

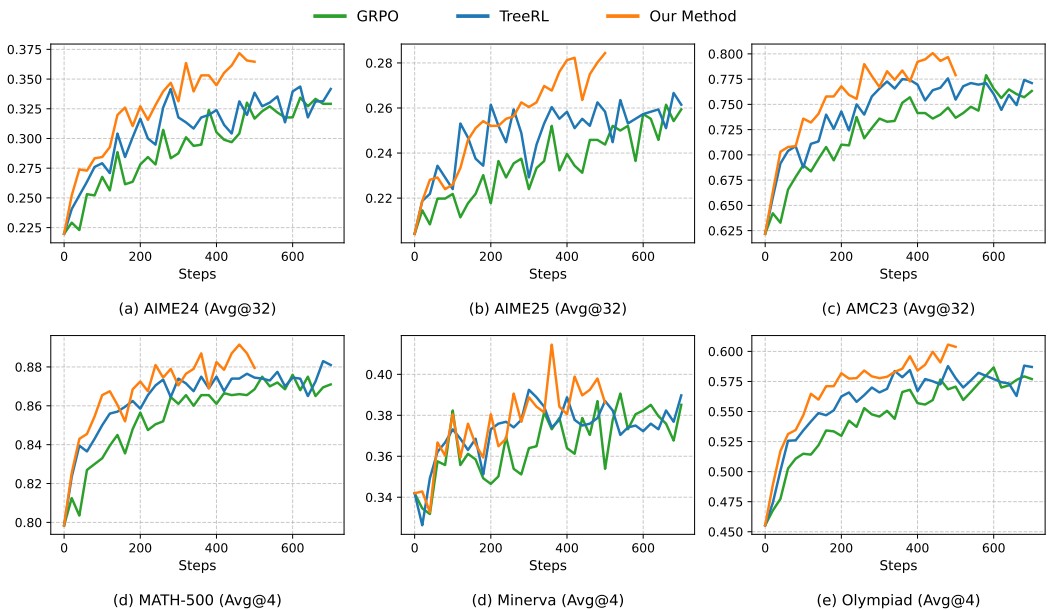

Figure 9: Test curves of GRPO, TreeRL, and our method on six benchmarks.

## C    CASES

The full response of Figure 2 is listed in Table 5 and steps with high FCI scores are shown in red.

Table 5: Full response.

> Okay, so I need to figure out how many positive three-digit integers are divisible by both 11 and 5. Hmm, let's see. I remember that if a number is divisible by both 11 and 5, it must be divisible by their least common multiple, right? But wait, I should double-check that because sometimes numbers share common factors, so maybe I need the least common multiple correctly.
> First, let me recall that 11 is a prime number, and 5 is also a prime number. Since they are both prime and different, the least common multiple (LCM) of them should just be their product. So, LCM(11, 5) = 11 * 5 = 55. That means we're looking for three-digit numbers divisible by 55.
> Alright, so the next step is to find out how many three-digit numbers are divisible by 55. Let's figure out the smallest and largest three-digit numbers that are divisible by 55. The smallest three-digit number is 100, and the largest is 999.
> I need to find the smallest three-digit number that's divisible by 55. So, let me divide 100 by 55 to see if it's a whole number. 55 * 1 = 55, 55 * 2 = 110. So, 55 * 2 = 110 is the first number

above 100 that is divisible by 55. Is that right? Let me divide 100 by 55 to find how many times 55 goes into 100. 100 / 55 is approximately 1.818. So, yes, the first multiple is 55 * 2 = 110. But 110 is the first three-digit number?

Wait, hold on. What's 55 * 1? That would be 55, which is not a three-digit number. So, we have to start at 55 * 2. So, the first multiple is 110, which is 55 * 2. But do we need to check if 55 * 1 is two-digit, which it is, so yes, we have to skip that.

Now, for the upper limit, the largest three-digit number is 999. So, let's divide 999 by 55 to get how many times 55 goes into 999. So, 999 / 55 is equal to approximately 18.16. So, 55 * 18 = 990, and 55 * 19 = 1045. Wait, 1045 is a four-digit number, which is too big. So, the largest multiple within three digits is 990, which is 55 * 18.

Wait, no—actually, 55 times 18 is 990, which is still less than 999. So, 18 is okay. Wait, but 55 * 19 is 1045, which is more than 999, so 990 is indeed the largest three-digit multiple.

So, our sequence of three-digit numbers divisible by 55 goes from 110 up to 990, with a common difference of 55 each time. So, to find the number of terms in this sequence, we can use the formula for the number of terms in an arithmetic sequence:

Number of terms = (Last term - First term) / Common difference + 1

So plugging in the numbers:

First term, a1 = 110

Last term, an = 990

Common difference, d = 55

So, (990 - 110) / 55 + 1.

Let me compute that.

First, calculate the difference: 990 - 110 = 880.

Then, divide by 55: 880 / 55 = 16.

Then, add 1: 16 + 1 = 17.

Therefore, there are 17 three-digit numbers divisible by both 11 and 5.

Wait, let me verify that. Maybe list a few and count.

Starting from 110,

110 (1st term) 165 (2nd term) 220 (3rd term) 275 (4th term) 330 (5th term) 385 (6th term) 440 (7th term) 495 (8th term) 550 (9th term) 605 (10th term) 660 (11th term) 715 (12th term) 770 (13th term) 825 (14th term) 880 (15th term) 935 (16th term) 990 (17th term)

Hmm, so that's 17 terms. Looks correct. So, the number is 17.

Alternatively, maybe using the formula with the general term:

The n-th term of the sequence is given by a_n = a1 + (n - 1)d.

So, since the last term is 990, let's set 990 = 110 + (n - 1)*55.

Subtract 110: 880 = (n -1)*55.

Divide by 55: 880 /55 = 16.

So, n -1 =16. Therefore, n=17.

Yes, same result. Seems solid.

Therefore, the number of positive three-digit integers divisible by both 11 and 5 is 17.

**Final Answer**

The number of positive three-digit integers divisible by both 11 and 5 is $\boxed{17}$. </think>

To determine how many positive three-digit integers are divisible by both 11 and 5, we first find the least common multiple (LCM) of 11 and 5. Since 11 and 5 are both prime, their LCM is $11 \times 5 = 55$. Therefore, we need to find the three-digit integers divisible by 55.

The smallest three-digit number divisible by 55 is found by dividing 100 by 55, which gives approximately 1.818. The next whole number is 2, so the smallest term is $55 \times 2 = 110$.

The largest three-digit number is 999. Dividing this by 55 gives approximately 18.16. The largest whole number is 18, so the largest term is $55 \times 18 = 990$.

To find the number of terms in this sequence, we use the formula for the number of terms in an arithmetic sequence:

$$\text{Number of terms} = \left( \frac{\text{Last term} - \text{First term}}{\text{Common difference}} \right) + 1$$

Substituting the values, we get:

$$\text{Number of terms} = \left(\frac{990 - 110}{55}\right) + 1 = \left(\frac{880}{55}\right) + 1 = 16 + 1 = 17$$

Thus, the number of positive three-digit integers divisible by both 11 and 5 is $\boxed{17}$.

## D  THE USE OF LARGE LANGUAGE MODELS

In this paper, LLMs are only used to polish the draft of some paragraphs, improving the clarity of the sentences. The key ideas, figures, and experiments are all from the human authors' contributions without the help of LLMs.

## E  ADDITIONAL DETAILS ON PROCESS-SUPERVISED RL

In this section, we provide additional preliminaries about process-supervised RL, especially TreeRL (Hou et al., 2025).

**Tree Constuction in TreeRL.**  TreeRL contains two-step sampling processes, including initial sampling and Monte Carlo sampling. For initial sampling, TreeRL samples 6 responses for each prompt. Starting from the prompt, we now have a tree with depth 1 and 6 leaf nodes. Then, TreeRL branches at the 2 tokens with the highest entropy for each response and sample 2 times at each branching point. After the branching, the tree has a depth of 3 and 30 leaf nodes (6 responses + 6 responses × 2 branching points × 2 samples). The responses at these leaf nodes are used for process reward estimation and policy training.

