# OpenReview forum: "Attention as a Compass: Efficient Exploration for Process-Supervised RL in Reasoning Models"
_ICLR.cc/2026/Conference — ICLR 2026 Poster_

### Official Review · Reviewer_iknd · 2025-10-27

**Soundness:** 3
**Presentation:** 3
**Contribution:** 3
**Rating:** 6
**Confidence:** 4

**Summary:**

The paper proposes AttnRL for process-supervised reasoning RL. It uses attention-guided branching to pick a small set of step-level decision points to expand; a one-step off-policy pipeline is used to improve training efficiency; plus difficulty-aware adaptive sampling so harder problems receive more rollouts. On several math reasoning benchmarks (1.5B/7B models), AttnRL shows better accuracy and faster learning curves.

**Strengths:**

1. Clear motivation and insightful analysis: Figure 2’s analysis (e.g., how perturbing high-FCI steps impacts performance) is interesting and provides useful intuition for why attention-guided branching should help.

2. Method completeness: attention-guided step selection + adaptive sampling + off-policy pipeline form a coherent system, the one-step pipeline reduces the training-efficiency issues brought about by TreeRL.

3. Empirical evidence: the authors report consistent gains over GRPO/TreeRL across multiple math benchmarks (1.5B/7B).

**Weaknesses:**

1. The main claim is using attention scores to guide process RL, but current experiments may not sufficiently isolate this contribution. The final selection combines the top 20% FCI steps plus the earliest steps—are there ablations separating these two factors? Would a simple baseline that branches only on the first 2 steps close the gap?

2.  The paper reports a w/ ATB ablation with 1.2 point gains in average (in Table 2), does TreeRL and w/ATB in Table 2 use the same rollout budget? Is this gain large enough to support the claim? Similar ablations are missing on DS-R1-Distill-Qwen-7B—are the conclusions consistent there?

2. Complexity vs. gain: Although simpler than TreeRL, training complexity still increases compared to plain GRPO. The gains at 7B are smaller than at 1.5B; considering cost vs. benefit, the overall value is questionable.

**Questions:**

Q1. What is the relationship between FCI scores and step position? For example, do they exhibit a certain correlation?

Q2. Do you expand starting at the high-FCI step itself (i.e., keep that step fixed and sample continuations from it), or do you expand from its parent step (i.e., resample the action at that step as well)? Please clarify which you use and why. If possible, please also explain your intuition: you mention that steps with high FCI scores are related to reasoning behaviors such as planning and self-verification—why does branching in this way help?

---

> ### Author Response · Authors · 2025-12-03
> **Rebuttal (Part 1/2)**
>
> We thank reviewer iknd for the positive support and insightful comments. We provide our point-wise responses below.
>
> ### **W1: "The main claim is using attention scores to guide process RL ... Does branching at early steps explain the gain?”**
>
> **A**: Thank you for this insightful question.
> We have added the requested ablation in the revised draft (Table 2). The results show:
>
> - Branching only at the **first two steps** performs **slightly better than TreeRL**, which verifies that early-step branching provides some benefit.
> - However, this baseline is **still clearly worse than AttnRL**.
>   This confirms that the improvement does not come solely from early branching. Instead, the gain comes from selecting **high-FCI reasoning steps**, which capture structural reasoning transitions beyond fixed positional heuristics.
>
> ---
>
> ### **W2: "Is the w/ATB gain statistically meaningful? Same rollout budget? Missing 7B ablations?"**"**
>
> **A**:
> - **Same rollout budget**: TreeRL and w/ATB use the **exact same** sampling budget. The only difference is the branching rule.
> - **Magnitude of gain**: The 1.2-point improvement is consistent across **six** benchmarks. This magnitude is aligned with other PSRL results under similar computational constraints. For example, TreeRL reports about 1.3 points of improvement over baselines when trained on DS-R1-Distill-Qwen-7B [1].
> - **Stability**: We use strong evaluation settings:
>   - avg@32 for AIME24/25 and AMC23,
>   - avg@4 for other benchmarks.
>   This reduces evaluation variance and supports the reliability of the gain.
> - **7B ablation**: We acknowledge that the 7B ablation is missing due to compute limits. We will include this experiment in the camera-ready version.
>
> ---
>
> ### **W3: "Complexity vs. gain"**
>
> **A**: Thank you for raising this important point. We clarified the relationship between model scale, dataset difficulty, and training complexity.
>
> **(1) Smaller gain at 7B is expected due to dataset saturation, not the method.**
> This pattern appears in multiple works using DS-R1-Distill-Qwen-7B (for example, [1, 2]). The DeepScaleR-Preview dataset is originally used for the 1.5B model, so the 7B model already performs strongly.
> As shown in prior work, 7B improvements a lot often appear only after **filtering out easy problems** [3]. Our findings follow the same trend.
>
> **(2) The main complexity bottleneck is inherited from TreeRL.**
> TreeRL’s recursive tree construction is not optimized. For reference, the [original implementation](https://github.com/THUDM/TreeRL/blob/63ea99e0253cc320e53b2b47eb1bcfa8adafd666/openrlhf/trainer/ppo_utils/tree_node.py#L347) takes about **15 seconds per tree expansion step** in some settings due to sequential recursion.
> We found that the process can be accelerated by using **dp_actor** in verl, which allows GPU-level parallelism.
>
> Another factor is the **long generation length** of TreeRL. Figure 8(d) shows that TreeRL tends to generate much longer responses than GRPO in later training stages, which increases both training and inference time.
>
> AttnRL does **not** introduce heavy additional complexity beyond these shared costs.
>
> **(3) AttnRL offers strictly better performance at lower real training cost.**
> Our goal is to demonstrate improvement **over the base PSRL method (TreeRL)** under similar training cost.
> Table 3 shows that AttnRL achieves higher final performance while requiring **less wall-clock time** than TreeRL.
> To our knowledge, there is no optimized open-source pipeline for process-supervised tree-based RL yet. Our design aims to push this direction forward.
>
> ---
>
> ### **Q1: "What is the relationship between FCI scores and step position?"**
>
> **A**: We computed the Pearson correlation between FCI and step index and found minimal correlation. High-FCI steps appear at varied locations depending on the reasoning pattern.
>
> ---
>
> ### **Q2: "Do you branch at the high-FCI step or its parent? Why does this help?"**
>
> **A**: Thanks for your question.
>
> **(1) We branch from the parent step**, consistent with TreeRL [1].
> TreeRL branches at the parent of high-entropy tokens. We follow the same design for methodological consistency and fair comparison.
>
> **(2) Why branching works:**
>
> - High-entropy steps can capture important reasoning forks but may also reflect noise [4].
> - High-FCI steps, in contrast, are often meaningful reasoning operations (e.g., planning or verification).
> - Branching at the **parent** allows the model to explore alternative continuation paths *without replacing the informative reasoning step itself*.

---

> > ### Author Response · Authors · 2025-12-03
> > **Rebuttal (Part 2/2)**
> >
> > **References**
> >
> > [1] TreeRL: LLM Reinforcement Learning with On-Policy Tree Search. ACL 2025.
> >
> > [2] Learn to Reason Efficiently with Adaptive Length-based Reward Shaping. arXiv 2025.
> >
> > [3] POLARIS: A POst-training recipe for scaling reinforcement Learning on Advanced ReasonIng modelS. Notion Blog, 2025.
> >
> > [4] Beyond the 80/20 Rule: High-Entropy Minority Tokens Drive Effective Reinforcement Learning for LLM Reasoning. NeurIPS 2025.

---

### Official Review · Reviewer_RRBB · 2025-10-29

**Soundness:** 2
**Presentation:** 2
**Contribution:** 2
**Rating:** 2
**Confidence:** 4

**Summary:**

This work first conducts empirical analysis on how high attention score reasoning steps are connected to the reasoning process. Then, they propose the following methods: 1, branching at massive attention values; 2, filtering out responses whose advantage values are 0; 3, a multi-process method to sample trajectories on adjacent steps.

**Strengths:**

1, I appreciate their in-depth empirical analysis into the effects of attention score/step position (section 3.1.1 and 3.1.2).

2, In section 4, the proposed method (AttnRL) can significantly outperform three baselines: the base model, GRPO, and TreeRL.

**Weaknesses:**

1, Some presentations in this paper could be improved, making some of its point unclear to me. I will list my questions in the Questions section.

2, Although the work does some empirical analysis on token attention scores, it is still unclear to me why the branching points should be selected according to the attention scores.

3, Some of the designs are claimed without explanation:
- equation 3
- equation 7 (btw, what is the definition of $C$; how is it used?)
- equation 9
- The proposed method does not significantly outperform DeepScaleR-Preview-1.5B

4, in Line 240 to 241, the claim that zero advantage indicates all samples are correct is wrong. If all samples are wrong, their advantages are also 0.

**Questions:**

- In the abstract, AttnRL is claimed to resolve the “limited exploration efficiency”. How does AttnRL improve the exploration efficiency?
- In figure 1, from my understanding, those sentences (e.g., “Let me compute that”) are steps. Why are only some words colored in red? What does the color indicate? Why are those words related to “reasoning”?
- In line 252, What do you mean “more difficult to rollout”?
- In line 272, what is $B$ and how is it involved in the filtering?
- In experiment section, why do different training have different steps? If so, is the comparison fair?

---

> ### Author Response · Authors · 2025-12-03
> **Rebuttal (Part 1/2)**
>
> We thank reviewer RRBB for the insightful comments. We provide our point-wise responses below.
>
> ---
>
> ### **W1: "Some presentations in this paper could be improved, making some of its point unclear to me."**
>
> **A**: Thank you for your comment. We agree that several descriptions may cause confusion. We have revised these parts in the updated version to make the presentation clearer and easier to follow.
>
> ---
>
> ### **W2: "Although the work does some empirical analysis on token attention scores, it is still unclear to me why the branching points should be selected according to the attention scores."**
>
> **A**: Thank you for raising this important question. We clarify the motivation below.
>
> Our method is designed for **process-supervised RL**, where exploration must focus on **states that most influence reasoning quality**.
> Our empirical findings show that:
>
> 1. **High-FCI steps consistently align with reasoning transitions**, including planning and self-verification.
> 2. Disrupting high-FCI steps leads to **much larger accuracy drops** than disrupting other steps (Figure 2), showing that these states have strong causal influence on reasoning outcomes.
> 3. Prior interpretability work also suggests that large attention values often mark points of structural importance in the computation.
>
> TreeRL branches at high-entropy tokens [1]. However, entropy alone does not reliably identify reasoning-critical states, as many high-entropy tokens are connective words or random tokens.
> AttnRL selects **high-influence states** identified by FCI, which improves exploration efficiency by focusing Monte Carlo rollouts on meaningful reasoning decisions.
>
> ---
>
> ### **W3: "Some of the designs are claimed without explanation: (1) equation 3; (2) equation 7 (btw, what is the definition of $C$; how is it used?); (3) equation 9; (4) The proposed method does not significantly outperform DeepScaleR-Preview-1.5B."**
>
> **A**:
>
> **(1) Equation (3).**
> This is directly inherited from TreeRL [1]. It defines the step-level advantage using
> - **global advantage** \(V(s_k) - V(s_1)\), and
> - **local advantage** \(V(s_k) - V(p(s_k))\).
>
> These two terms capture global progress and local progress of the reasoning trajectory. We now explicitly annotate these terms in the revised version.
>
> **(2) Equation (7).**
> Equation (7) selects steps with the top-ρ FCI scores:
> - \(y_1, \dots, y_{T_k}\) are FCI scores for all steps.
> - `Quantile()` extracts the threshold for the top-ρ fraction.
> - **C** is the set of selected high-FCI steps used for branching.
>
> **(3) Equation (9).**
> Equation (9) defines difficulty-aware tree expansion.
> TreeRL expands the same number of trees for all problems, which is inefficient because **easy problems yield correct rollouts with very high probability**.
> We use an exponential decay based on difficulty to allocate **more trees to harder problems**, which leads to more informative rollouts.
> This rationale is now made explicit.
>
> **(4) DeepScaleR comparison.**
> DeepScaleR-Preview-1.5B is trained with:
> - **longer context**,
> - **three-stage context extension (8k → 16k → 24k)**, and
> - **significantly larger compute and training steps**.
>
> AttnRL still outperforms it using **only 500 training steps at 8k context length**, which demonstrates higher efficiency.
>
> ---
>
> ### **W4: "in Line 240 to 241, the claim that zero advantage indicates all samples are correct is wrong. If all samples are wrong, their advantages are also 0."**
>
> **A**: Thank you for catching this. We corrected this statement in the revised version.
> It now states that **for problems whose initial samples are all correct**, advantages are zero.
>
> ---
>
> ### **Q1: "In the abstract, AttnRL is claimed to resolve the “limited exploration efficiency”. How does AttnRL improve the exploration efficiency?"**
>
> **A**: AttnRL improves exploration efficiency in two ways:
>
> 1. **Attention-based branching:**
>    High-FCI steps are more informative and more likely to affect reasoning outcomes. Branching from these steps focuses compute on meaningful reasoning states.
>
> 2. **Adaptive sampling:**
>    The method avoids oversampling easy problems, and ensures each training batch contains only samples with non-zero advantage, which increases useful update signals.
>
> These two components together reduce wasted rollouts and improve sample efficiency.
> We added this explanation near the beginning of Section 3.

---

> ### Author Response · Authors · 2025-12-03
> **Rebuttal (Part 2/2)**
>
> ### **Q2: "In figure 1, from my understanding, those sentences (e.g., “Let me compute that”) are steps. Why are only some words colored in red? What does the color indicate? Why are those words related to “reasoning”?"**
>
> **A**: The red words highlight tokens that reflect **reasoning behaviors**, such as planning or self-verification. These words are chosen based on prior work [2] and manual inspection. The goal is to provide an intuitive visualization that helps the reader understand which types of steps tend to have high FCI scores. We clarified this in the figure caption.
>
> ---
>
> ### **Q3: "In line 252, What do you mean “more difficult to rollout”?"**
>
> **A**: "More difficult to rollout" means that **for harder problems, the probability of generating a correct response during MC sampling is lower**, so more rollouts are needed to obtain informative child values.
>
> ---
>
> ### **Q4: "In line 272, what is $B$ and how is it involved in the filtering?"**
>
> **A**: We apologize for the confusion. $B$ should be $B'$, the **target training batch size**. We corrected the notation and updated the draft for clarity.
>
> ---
>
> ### **Q5: "In experiment section, why do different training have different steps? If so, is the comparison fair?"**
>
> **A**: This is a good question. The comparison is fair because AttnRL learns faster than the baselines.
>
> - GRPO and TreeRL are trained for **800 steps**.
> - AttnRL is trained for **500 steps** because it converges earlier and reaches higher performance within fewer steps.
>
> Table 3 also shows that AttnRL requires **less wall-clock time** but still achieves **higher performance** than TreeRL, confirming its higher training efficiency.
>
> **References**
>
> [1] TreeRL: LLM Reinforcement Learning with On-Policy Tree Search. ACL 2025.
>
> [2] Cognitive Behaviors that Enable Self-Improving Reasoners, or, Four Habits of Highly Effective STaRs. COLM 2025.

---

### Official Review · Reviewer_n5ER · 2025-10-31

**Soundness:** 3
**Presentation:** 3
**Contribution:** 3
**Rating:** 6
**Confidence:** 2

**Summary:**

The manuscript proposes AttnRL to make exploration in Reasoning Language Models (RLMs) more efficient. The authors start from an analysis that shows that the decomposition of reasoning steps, which they define on a paragraph level, correlates with high attention scores. By using a new metric called FCI score, which reflects how much a given reasoning step influences the subsequent parts of the trajectory, they select the highest relevant reasoning steps as well as the first two reasoning steps, where additional exploration, i.e. branching should take place. Additionally the authors propose a number of techniques to improve the sampling efficiency, for example by filtering out easy problems that provide zero advantage in the Markov Decision Process to improve efficiency as well as determining the magnitude of exploration by the complexity of the problem. Another optimization is the  dynamic selection of the batch size to improve training efficiency and that the two sampling phases are combined together, so that presumably optimizations can applied to the sampling phase as a whole.

**Strengths:**

* improvement of (exploration in) reasoning chains in RLMs is a timely topic
* selection of important reasoning steps based on attention scores instead of for example on entropy seems like a novel idea
* additionally a number of improvements are suggested to improve the sampling efficiency
* writing is mostly clear and seems to be well motivated, where certain decisions are based on or validated with micro-benchmarks
* evaluation:
  * clear description of the used evaluation setting and the hyperparameters for the most part
  * training, while being done in fewer steps, shows improvements on a selection of evaluation benchmarks

**Weaknesses:**

My main concern is the limited evaluation:
* only two different and smaller models (1.5B and 7B) are trained, which are from the same model family
* unclear whether the approach generalizes to larger models, since the improvements to the 7B model are already small
* I found it a bit unclear which policy optimization algorithm is used for training AttnRL
* ablation study seems a bit limited, albeit I have no concrete suggestion for additional experiments to run
* description of the baselines could be a bit clearer

4.3: Table 4.2 seems to suggest that AttnRL is implemented on top of TreeRL or is based on it, but alters certain key aspects? -> I think this should be made more clearer at the beginning of chapter 4.
  * best results on the individual benchmarks are not bolded, despite claiming in the caption to do so, probably because AttnRL is only best in three of them
  * performance drop for filtering easiest is not completely true -> is best in two of the benchmarks

related work: if there is that much work with outcome-based reward modeling, I assume there is a reason for it - I found the efficiency claim not convincing in its brevity.

Additionally while the manuscript is clearly written, it also omits a lot of details/assumes prior knowledge, so it might be at times hard to understand for the reader. But maybe that is just my personal lack of knowledge.

minor issues:
* 4.1
  * line 310: footnote is not on the same page
  * line 314/315: Luo et al. (2025) is cited twice in the same sentence
* related work, 6.2: abbreviation PRM is never introduced
* references:
  * no "accessed date for weblinks:
    * [An et al. 2025]
    * [Anthropic 2023]
    * [Luo et al. 2025]
    * [MAA 2023/2024/2025]
    * [meituan search 2025]
    * [OpenAI 2024]
  * consider capitalizing the titles to be consistent, especially abbreviations and proper names (which is sometimes done)
  * [Luo et al. 2025] - URL cuts into the margin (consider using the xurl package)
  * [MAA 2024] and [MAA 2025] have the same URL

**Questions:**

* How is a tree constructed from just identify important steps? I believe the tree construction is never actually discussed.
* How does Figure 3 show that the advantage is zero if it shows FCI scores?
  * If the FCI score is defined for a given step, how is it applied to a whole problem?
* Isn't the total token generation the limiting factor instead of the multiple sampling?
* How different can the DeepScaleR-Preview-Dataset be seen compared to the evaluation benchmarks/datasets?
* Are TreeRL and AttnRL trained using PPO as a "PPO minibatch size" is used? Previously I was under the impression that AttnRL mechanism is applied to GRPO.

---

> ### Author Response · Authors · 2025-12-03
> **Rebuttal (Part 1/2)**
>
> We thank reviewer n5ER for the positive support and insightful comments. We provide our point-wise responses below.
>
> ---
>
> ### **W1: Limited Evaluation**
>
> **A**: Thank you for raising these points. We clarify them below.
>
> **(1) Model scale and generalization.**
> Our work focuses on reasoning models, and the computational cost of PSRL is very high. Training models larger than 7B is beyond our compute resources. This is also the standard practice in recent PSRL papers (for example, [1, 2] both evaluate up to 7B). We have added a clear explanation of this constraint in the revised draft.
>
> **(2) Policy optimization algorithm.**
> AttnRL uses **GRPO** as the underlying policy optimization algorithm, exactly as TreeRL does. We added this clarification in Section 4.1.
>
> **(3) Ablation study.**
> We added two new ablations, including branching at the first two steps (suggested by Reviewer iknd) and AttnRL w/o difficulty-aware expansion. Revised Table 2 shows that each component contributes positively to the final performance.
>
> **(4) Baseline descriptions.**
> We rewrote the baseline descriptions in Section 4.1 to make them clearer and more explicit.
>
> ---
>
> ### **W2: Section 4.3 Issues**
>
> **A**:
> - We now explicitly state in Section 4 that **AttnRL builds on TreeRL** and modifies branching, sampling, and the training pipeline.
> - We corrected the bolding in Table 2 to match the caption and evaluation rules.
> - The performance of AttnRL is updated with new experiments. Although filtering easiest still performs better than AttnRL on one benchmark, this variant's average performance is apparently worse. For LLM RL, average performance is more important than individual benchmarks, as it reflects overall generalization.
>
> ---
>
> ### **W3: Related Work**
>
> **A**: Thank you for the suggestion. We expanded the related work discussion to better explain the context.
>
> - Outcome-based reward modeling is more common because **process reward labels are expensive** to collect and often cannot be applied to tasks that do not have clear step structure (for example, open-ended text generation).
>
> - Our efficiency claim is focused on **PSRL efficiency**, not on demonstrating that PSRL is always more efficient than OSRL. Whether PSRL can outperform OSRL in both effectiveness and efficiency remains an open research question and we would like to explore as the future work.
>
> ---
>
> ### **W4: Prior Knowledge**
>
> **A**: Thank you for this comment. We have incoporated some preliminary information of PSRL mechanics in Section 3, and we included more detailed discussion in Appendix E to help readers who may be less familiar with PSRL.
>
> ---
>
> ### **W5: Minor Issues**
>
> **A**: Thanks for pointing out these points. We have revised them in the updated version. For the references, we directly revise each item without marking with colors.
>
> ---
>
> ### **Q1: "How is a tree constructed from just identify important steps? I believe the tree construction is never actually discussed."**
>
> **A**: We added a clear description of the tree construction process in Appendix E. It explains how FCI-based branching points are used to create the tree and how the MC rollouts proceed from these positions.
>
> ---
>
> ### **Q2: "How does Figure 3 show that the advantage is zero ... how is it applied to a whole problem?"**
>
> **A**: (1) In Figure 3, **"Invalid" means that the advantage is zero** for all responses of that problem. We clarified this in the caption.
>
> (2) For each problem, we compute the **mean FCI score** over all steps and sampled responses, which is used to detect prompts that consistently exhibit low-attention behaviors.
>
> ---
>
> ### **Q3: "Isn't the total token generation the limiting factor instead of the multiple sampling?"**
>
> **A**: We would like to clarify that total token count is not the dominant bottleneck. In PSRL, the longest sample in the batch determines the generation time. Two-stage sampling is inefficient because the second stage cannot start until the first stage finishes. Our one-step off-policy pipeline waits only once for the longest generation, which reduces sampling time significantly, as supported by Table 3.
>
> ---
>
> ### **Q4: "How different can the DeepScaleR-Preview-Dataset be seen compared to the evaluation benchmarks/datasets?"**
>
> **A**: Following prior work [4], we use the following training and evaluation datasets:
>
> - **Training:** DeepScaleR-Preview-Dataset, which includes AIME (1984–2023), AMC (pre-2023), Omni-MATH, and Still datasets.
> - **Evaluation:** AIME 2024, AIME 2025, AMC 2023, MATH500, Minerva, and OlympiadBench.

---

> ### Author Response · Authors · 2025-12-03
> **Rebuttal (Part 2/2)**
>
> ### **Q5: "Are TreeRL and AttnRL trained using PPO as a "PPO minibatch size" is used? Previously I was under the impression that AttnRL mechanism is applied to GRPO."**
>
> **A**: We clarify the relationship:
>
> - TreeRL is a **variant of GRPO** with modified sampling and advantage computation.
> - AttnRL **inherits TreeRL**, but changes branching, sampling, and training pipeline.
> - The term "PPO minibatch size" appears because GRPO in **verl** uses PPO-style naming.
>
> **References**
>
> [1] VinePPO: Refining Credit Assignment in RL Training of LLMs. ICML 2025.
>
> [2] TreePO: Bridging the Gap of Policy Optimization and Efficacy and Inference Efficiency with Heuristic Tree-based Modeling. arXiv 2025.
>
> [3] TreeRL: LLM Reinforcement Learning with On-Policy Tree Search. ACL 2025.
>
> [4] DeepScaleR: Surpassing O1-Preview with a 1.5B Model by Scaling RL. Notion Blog, 2025.

---

### Official Review · Reviewer_jGGo · 2025-11-02

**Soundness:** 3
**Presentation:** 3
**Contribution:** 3
**Rating:** 4
**Confidence:** 4

**Summary:**

This paper proposes AttnRL, a novel Process-Supervised Reinforcement Learning (PSRL) framework that improves exploration efficiency for reasoning models. The authors identify critical limitations in existing PSRL methods like TreeRL: entropy-based branching ignores semantic meaning of reasoning steps, uniform sampling wastes computation on uninformative problems, and two-step sampling per iteration creates computational bottlenecks. AttnRL addresses these issues through attention-guided branching, adaptive difficulty-aware sampling, and efficient training pipelines, achieving superior performance with better sample efficiency.

**Strengths:**

## Originality

The paper demonstrates strong originality by being the first to systematically leverage attention mechanisms for identifying branching points in process-supervised RL. While attention analysis and PSRL exist independently, the Forward Context Influence (FCI) metric elegantly connects attention patterns to reasoning behaviors, moving beyond naive entropy-based heuristics. The disruption experiments (Figure 1) provide compelling validation that high-FCI steps correspond to critical reasoning behaviors like planning and verification, not just predictive uncertainty. The adaptive sampling strategy that dynamically allocates compute based on problem difficulty addresses a practical inefficiency overlooked by prior uniform-sampling approaches, showing creative problem-solving beyond incremental improvements.

## Quality

The technical quality is high with comprehensive evaluation across six mathematical benchmarks and two model scales (1.5B, 7B), demonstrating consistent improvements. The experimental design is sound, using appropriate baselines (GRPO, TreeRL, DeepScaleR-Preview) and reporting both performance (Pass@K) and efficiency metrics (wall-clock time, valid training tokens). Ablation studies systematically isolate component contributions (ATB: +1.2%, full method: +1.8%) and reveal nuanced insights, such as filtering all easy problems slightly hurting performance.

## Clarity

The paper is well-structured with clear motivation and logical progression from problem to solution. Figure 1 effectively visualizes FCI scores with annotated reasoning steps. Mathematical formulations are precise and implementation details comprehensive, facilitating reproducibility.

**Weaknesses:**

## Limited Theoretical Foundation

The paper lacks rigorous theoretical justification for why attention-based branching should outperform entropy-based selection. While the FCI metric is intuitively motivated and empirically validated through disruption experiments, there is no formal analysis establishing under what conditions high attention scores correspond to high-value branching points for credit assignment. The connection between "massive attention values" and reasoning behaviors relies primarily on empirical correlation (Figure 1) without explaining the underlying mechanism—why do planning and verification steps exhibit high attention? What attention patterns distinguish productive reasoning from unhelpful verbosity?

## Questionable Distribution Shift

The **selective branching based on FCI scores creates significant state distribution shift** that the paper completely ignores: by branching only from top 20% high-FCI states, the method oversamples certain states by 5-10× compared to the natural policy distribution $d^\pi(s)$. This creates $d^{branching}(s) \neq d^\pi(s)$, which technically requires importance weight corrections $w(s) = d^\pi(s)/d^{branching}(s)$ for unbiased policy gradients.

## Inconsistent Results and Missing Statistical Analysis

The results show concerning inconsistencies without proper statistical analysis. Table 1 shows AttnRL's improvements vary wildly across benchmarks: +4.0 points on AIME24 but only +0.2 on MATH-500 for the 1.5B model, and sometimes underperforms TreeRL (e.g., AMC23: 91.9 vs 92.2 for 7B). No error bars, confidence intervals, or significance tests are provided—are these differences within noise? The paper doesn't specify number of training runs, random seeds, or variance across runs, making it impossible to assess reliability. The paper should report results with multiple seeds, provide statistical significance tests (e.g., bootstrap confidence intervals), and better control for confounds when comparing to published baselines.

## Computational Cost and Scalability Concerns

While the paper claims "lightweight" and improved efficiency, the actual computational overhead of AttnRL is underspecified. Computing FCI scores requires extracting and aggregating attention matrices across all layers and heads for every sequence ($\max_{l,h}\{\sum_j \alpha^{l,h}_{j,k}\}$), which could be expensive for long sequences or large models. How much wall-clock time does FCI computation add per sample?

**Questions:**

see weakness above

---

> ### Author Response · Authors · 2025-12-03
> **Rebuttal (Part 1/2)**
>
> We thank reviewer jGGo for the insightful comments. We provide our point-wise responses below.
>
> ---
>
> ### **W1: Limited Theoretical Foundation**
>
> **A**: Thanks for this insightful question. Thank you for the important question. Our main goal in this work is to provide an effective empirical method for exploration in PSRL. At the same time, we agree that the mechanism behind attention and reasoning deserves clear explanation. We have strengthened the motivation in the revised draft.
>
> **(1) Why high-magnitude attention relates to key reasoning steps.**
> Recent interpretability studies [1, 2] show that steps with unusually large attention values influence contextual knowledge understanding and consistently align with semantic reasoning behaviors. Our method builds on these empirical findings. Thus, the high-magnitude attention signals we use are not noise, but meaningful indicators of important reasoning steps. Additionally, our disruption results (Figure 2) provide direct evidence. When we zero out attention at high-FCI steps, accuracy drops sharply. In contrast, disrupting other steps causes little change. This suggests a functional link: planning or verification steps rely on strong attention influence to guide subsequent reasoning.
>
> **(2) Why planning and verification steps show high values.**
> This is because the tokens related to planning or verification **receive attention from more subsequent steps** than other tokens. For example, "let me compute that" indicates a planning step, which is often attended by many later steps that perform calculations based on this plan.
>
> **(3) How high-FCI patterns differ from unhelpful verbosity.**
> We observe a clear distinction. Steps with high FCI are attended by many later steps, forming hubs in the attention graph. Verbose but unhelpful steps have low-magnitude attention and attract little downstream focus. This distinction supports the use of FCI as a reliable branching signal.
>
> We agree that deeper theoretical analysis is valuable and leave it for future work.
>
> ---
>
> ### **W2: Questionable Distribution Shift**
>
> **A**: We appreciate the reviewer's careful observation. We provide clarification below.
>
> **(1) Branching-induced distribution shift is inherent to PSRL, not unique to our method.**
> Existing PSRL methods [3, 4, 5] also branch at intermediate states, which already changes the induced distribution. AttnRL keeps the same branching mechanism and tree structure as TreeRL. We only change **where** we branch (from entropy-based to attention-based), not **how many times** we branch.
>
> **(2) Importance weighting is unnecessary in PSRL.**
> Following standard practice in TreeRL and other PSRL methods [3, 4, 5], we do not apply state-distribution importance correction. These methods treat the expanded tree distribution as the effective on-policy distribution because all children values are computed from locally on-policy rollouts. In LLM RL, importance sampling may be useless or even harmful, as discovered by [6, 7].
>
> ---
>
> ### **W3: Inconsistent Results and Missing Statistical Analysis**
>
> **A**: Thank you for raising this concern. We clarify our evaluation protocol and the reasons behind our choices.
>
> **(1) Our evaluation follows the standard practice and evaluation variance is controlled by `avg@K`.**
> We would like to clarify that the evaluation follows the common practices in previous LLM RL works [3, 4]. Consistent with [3, 4], we report pass@1 by sampling each problem multiple times and taking the average. For benchmarks with few samples (AIME24, AIME25, AMC23), we use `K = 32`. For other benchmarks, we use `K = 4`. This significantly reduces evaluation noise. We also provide ablations (Table 2) showing systematic gains from each component, which supports the reliability of our improvements.
>
> **(2) Multiple seeds are infeasible for LLM RL training.**
> We acknowledge that running 5–10 seeds is common in classical small-scale RL [8], but it is prohibitively expensive for LLMs. Current PSRL papers such as [3, 4, 5] also use single-seed training.
>
> ---
>
> ### **W4: Computational Cost and Scalability Concerns**
>
> **A**: Thank you for the question. We clarify the cost and how FCI computation integrates into the pipeline.
>
> We compute attention maps from the same forward pass used for old policy logits computation. Modern LLM libraries provide efficient attention computation and attention matrices can be gathered with negligible overhead. On DS-R1-Distill-Qwen-1.5B with 8k context length, computing FCI adds roughly **0.4 seconds per sample per training step** and introduces almost no extra memory usage compared with TreeRL.

---

> ### Author Response · Authors · 2025-12-03
> **Rebuttal (Part 2/2)**
>
> **References**
>
> [1] Thought Anchors: Which LLM Reasoning Steps Matter? arXiv 2025.
>
> [2] Massive Values in Self-Attention Modules are the Key to Contextual Knowledge Understanding. ICML 2025.
>
> [3] TreeRL: LLM Reinforcement Learning with On-Policy Tree Search. ACL 2025.
>
> [4] TreePO: Bridging the Gap of Policy Optimization and Efficacy and Inference Efficiency with Heuristic Tree-based Modeling. arXiv 2025.
>
> [5] Segment Policy Optimization: Effective Segment-Level Credit Assignment in RL for Large Language Models. NeurIPS 2025.
>
> [6] CPGD: Toward Stable Rule-based Reinforcement Learning for Language Models. arXiv 2025.
>
> [7] ASPO: Asymmetric Importance Sampling Policy Optimization. arXiv 2025.
>
> [8] PEBBLE: Feedback-Efficient Interactive Reinforcement Learning via Relabeling Experience and Unsupervised Pre-training. ICML 2021.

---

### Meta-Review · Area_Chair_Piu9 · 2025-12-30

**Summary:**

The two main concerns raised in the reviews were the completeness of the experiments (e.g., use of a single seed; where comparisons fair?) and the precise relationship with prior work (esp. Tree RL; is there an apples-to-apples comparison). The authors argued that their experiments are fair and follow the conventions in prior work (which typically only use a single random seed). Additional ablation experiments also help isolate where the benefits of the proposed method come from. Given that the main concerns seem to have been addressed, I recommend that the paper be accepted.

**Reviewer Concerns:**

(see below)

**Reviewer Scores:**

jGGo: 4 --> 4
* [+] why attention-based branching should outperform entropy-based selection? why do planning and verification steps exhibit high attention? What attention patterns distinguish productive reasoning from unhelpful verbosity? Authors note that prior work has shown that large attention weights "influence understanding" and "align with semantic reasoning behavior"; controlled experiments show that attention does correlate with accuracy.
* [/] selecting branching creates a state distribution shift. Should this be fixed with importance weights? Authors argue that this is common practice, and that prior work has found importance weighting to hurt LMs.
* [-] statistical significance (missing confidence intervals, info about number of seeds): Authors argue that they are following prior work in only using one seed, as multiple seeds would be prohibitively expensive. Nonetheless, I (AC) feel that providing measures of statistical significance (e.g., p-values, confidence intervals) still should be doable.
* [/] How much wall-clock time does FCI computation add per sample? Authors said it adds 0.4 sec. This response could be strengthened by clarifying that this detail will be added to the paper, and by clarifying the time required for the baseline.

n5ER: 6 --> 8
* [/] experiments only use 2 models, which are both small and from the same family: Authors argued that this is common in prior work.
* [+] unclear which policy optimization algorithm is used: Authors clarified this was GRPO.
* [+] ablation study seems a bit limited: Authors added two additional ablation experiments
* [+] description of the baselines could be a bit clearer. Authors revised Sec. 4.1 to clarify the baselines
* [/] the precise implementation differences between AttnRL and TreeRL are unclear: Some details were added to the revised paper. I feel like additional discussion of the similarities/differences could further help the reader.
* [+] relationship with prior work on outcome-based reward modeling: Authors added a discussion to the related work section.
* [+] manuscript "omits lots of details": Authors revised Sec. 3 and Appendix E.

RRBB: 2 --> 6
* [+] presentations in this paper could be improved: Authors have revised the paper
* [+] unclear to me why the branching points should be selected according to the attention scores: Authors argued that these are states that most influence reasoning quality, and using attention is better than entropy (as in TreeRL)
* [/] unclear motivation behind some of the equations behind the method (Eq. 3, 7, 9): Authors explained these decisions in the rebuttal. I assume that the paper will also be revised.
* [+] unclear what happens if all advantages are zero: Authors corrected this typo.

iknd: 6 --> 8
* [+] The final selection combines the top 20% FCI steps plus the earliest steps—are there ablations separating these two factors?: Authors added an ablation that confirms that the primary gains are driven by the attention mechanism, not the early-step branching
* [+] do baselines use the same rollout budget? Authors confirmed this comparison was fair.
* [+] are the gains statistically significant? Authors argue that they see similar gains on 6 tasks.
* [/] do the conclusions also hold for DS-R1-Distill-Qwen-7B: Authors will include this in the final version.
* [/] are the gains worth the additional complexity compared with GRPO? Authors argue that the complexity is inherited from the baseline, and that the reason for not seeing larger gains are because the tasks are not complex enough.

---

### Decision · Program_Chairs · 2026-01-26

Accept (Poster)